# Utility of Platelet Endothelial Cell Adhesion Molecule 1 in the Platelet Activity Assessment in Mouse and Human Blood

**DOI:** 10.3390/ijms22179611

**Published:** 2021-09-04

**Authors:** Natalia Marcinczyk, Tomasz Misztal, Anna Gromotowicz-Poplawska, Agnieszka Zebrowska, Tomasz Rusak, Piotr Radziwon, Ewa Chabielska

**Affiliations:** 1Department of Biopharmacy, Medical University of Bialystok, 15-222 Bialystok, Poland; anna.gromotowicz@umb.edu.pl (A.G.-P.); ewa.chabielska@umb.edu.pl (E.C.); 2Department of Physical Chemistry, Medical University of Bialystok, 15-089 Bialystok, Poland; tomasz.misztal@umb.edu.pl (T.M.); tomasz.rusak@umb.edu.pl (T.R.); 3Regional Centre for Transfusion Medicine, 15-950 Bialystok, Poland; azebrowska@rckik.bialystok.pl (A.Z.); piotr.radziwon@umb.edu.pl (P.R.); 4Department of Haematology, Medical University of Bialystok, 15-276 Bialystok, Poland

**Keywords:** PECAM-1, thrombosis, inflammation, platelet

## Abstract

In our previous study, we introduced the platelet endothelial cell adhesion molecule 1 (PECAM-1)/thrombus ratio, which is a parameter indicating the proportion of PECAM-1 in laser-induced thrombi in mice. Because PECAM-1 is an antithrombotic molecule, the higher the PECAM-1/thrombus ratio, the less activated the platelets. In this study, we used an extracorporeal model of thrombosis (flow chamber model) to verify its usefulness in the assessment of the PECAM-1/thrombus ratio in animal and human studies. Using the lipopolysaccharide (LPS)-induced inflammation model, we also evaluated whether the PECAM-1/thrombus ratio determined in the flow chamber (without endothelium) differed from that calculated in laser-induced thrombosis (with endothelium). We observed that acetylsalicylic acid (ASA) decreased the area of the thrombus while increasing the PECAM-1/thrombus ratio in healthy mice and humans in a dose-dependent manner. In LPS-treated mice, the PECAM-1/thrombus ratio decreased as the dose of ASA increased in both thrombosis models, but the direction of change in the thrombus area was inconsistent. Our study demonstrates that the PECAM-1/thrombus ratio can more accurately describe the platelet activation status than commonly used parameters such as the thrombus area, and, hence, it can be used in both human and animal studies.

## 1. Introduction

Platelet endothelial cell adhesion molecule 1 (PECAM-1) is a transmembrane glycoprotein that belongs to the immunoglobulin superfamily. It is present on the surface of platelets, endothelial cells, monocytes, neutrophils, and lymphocytes [1]. Previous animal studies have shown that PECAM-1 plays an important role in the regulation of the thrombotic process and that increased expression and activity of this protein results in the inhibition of thrombus formation [2]. It was also reported that PECAM-1 knockout mice exhibited a prothrombotic phenotype [3]. PECAM-1 plays a pivotal role in the regulation of platelet activity by inhibiting the activation of receptors bearing an immunoreceptor tyrosine-based activation motif (ITAM). One such receptor is glycoprotein VI Fc receptor γ-chain (GPVI/FcR γ), which is the main collagen receptor in platelets. During collagen binding, ITAMs located on the cytoplasmic tail of GPVI become phosphorylated by SFK kinases, providing docking sites for Syk kinases. In turn, Syk kinases phosphorylate the LAT signalosome, resulting in the activation of phosphoinositide 3-kinase (PI3K) and phospholipase C γ2 (PLCγ2). The signal transduction pathways triggered by PI3K and PLCγ2 activate platelets. An immunoreceptor tyrosine-based inhibition motif (ITIM) is located on the cytoplasmic tail of PECAM-1. ITIM is regarded as a key element in the antithrombotic activity of PECAM-1 by inhibiting the above pathways. ITIMs are phosphorylated by SFK kinases, which allows the binding of Src homology region 2 domain-containing protein tyrosine phosphatase 1 (Shp1). As a result of these motifs, Shp1 is located adjacent to phosphorylated ITAM and enables ITAM dephosphorylation, which inhibits the signaling pathway associated with platelet activation. The phosphorylation of ITIMs occurs after platelet activation or the clustering of PECAM-1 on the platelet surface [1]. PECAM-1 has also been shown to suppress ADP- and thrombin-induced platelet activation, but the precise molecular mechanism underlying this process is unknown [2].

Our previous study was the first to propose a parameter known as the PECAM-1/thrombus ratio for evaluating platelet activity. The PECAM-1/thrombus ratio indicates the proportion of PECAM-1 in a laser-induced thrombus in a mouse mesenteric vein. As PECAM-1 is considered an antithrombotic molecule, the higher the PECAM-1/thrombus ratio, the less activated the platelets in the thrombus. Using this ratio, we confirmed the antiplatelet activity of acetylsalicylic acid (ASA) [4] and enhanced the platelet activity in diabetic mice under intravital conditions [5]. These results revealed that the PECAM-1/thrombus ratio is a potentially useful parameter for assessing platelet activity in laser-induced thrombosis. Furthermore, the PECAM-1/thrombus ratio could be used to assess the activity of human platelets *ex vivo*. Therefore, in the present study, we used a model of thrombus formation on collagen fibers under controlled flow conditions (flow chamber model), which allowed the extracorporeal induction of thrombosis and the observation of the thrombotic process in human blood.

Immunothrombosis, a complex process triggered by inflammation, involves changes in the activity of platelets and innate immune cells and affects the functions of endothelial cells and the coagulation system [6]. In this study, we used a lipopolysaccharide (LPS)-induced mouse model of inflammation to evaluate platelet activity using the PECAM-1/thrombus ratio in two thrombosis models: a flow chamber model and laser-induced thrombosis model. The latter was applied for the intravital observation of the thrombotic process at the site of the mesenteric vein injury, where the thrombus forms within 80 s on a small area of exposed subendothelial matrix and is mainly composed of platelets. In the flow chamber model, the thrombus is formed ex vivo on collagen fibers and also predominantly consists of platelets. However, the main difference between these two models is the presence or absence of endothelium, which affects the process of thrombus formation.

The overriding goal of our study was to identify a parameter that allows the assessment of platelet activity in a thrombus (in vivo and ex vivo). Therefore, the first aim of our experiment was to establish whether the PECAM-1/thrombus ratio calculated in the flow chamber model (without endothelium) differs from that calculated in the intravital thrombosis model (with endothelium) and, hence, whether this parameter in combination with the flow chamber may be a reliable tool for the assessment of ex vivo platelet activity in pathological states such as inflammation. The second aim was to evaluate whether the PECAM-1/thrombus ratio is applicable to human blood. For these purposes, we used ASA, a compound exhibiting potent and well-described antiplatelet activity, as a reference [7].

## 2. Results

### 2.1. Laser-Induced Thrombosis in Mice (Intravital)

In mice treated with lipopolysaccharide (LPS) alone, the thrombus area was found to be increased (Figure 1a), but the PECAM-1/thrombus ratio was not altered (Figure 1b). In the mice treated with LPS as well as ASA at 10 and 30 mg/kg, the thrombus area decreased (Figure 1a), but the PECAM-1/thrombus ratio was unchanged (Figure 1b). An increased PECAM-1/thrombus ratio was observed only in the LPS + ASA 3 mg/kg group (Figure 1b).

### 2.2. Thrombus Formation in the Flow Chamber (Mice, Ex Vivo)

In control mice (mice without inflammation) treated with ASA (at 10 and 30 mg/kg), the thrombus area decreased (Figure 2a), and the PECAM-1/thrombus ratio increased (Figure 2b) in a dose-dependent manner. In LPS-treated mice (mice with inflammation), LPS alone did not change either the thrombus area or the PECAM-1/thrombus ratio (Figure 2c,d). In LPS-treated mice, treatment with ASA at all tested doses caused a reduction in the thrombus area (Figure 2c). However, in the LPS + ASA 10 mg/kg group, the thrombus area was increased in comparison to the LPS + ASA 3 mg/kg and LPS + ASA 30 mg/kg groups (Figure 2c). In LPS-treated mice, treatment with ASA also increased the PECAM-1/thrombus ratio; however, in contrast to the control mice, the PECAM-1/thrombus ratio decreased with the increase in the dose of ASA (Figure 2d).

### 2.3. Bleeding Time (BT) in LPS-Treated Mice

The BT was shortened in the LPS and LPS + ASA 3 mg/kg groups. In LPS-treated mice, prolonged BT was observed only after treatment with ASA at a dose of 10 mg/kg, whereas ASA administered at 30 mg/kg did not affect BT in LPS-treated mice (Figure 3).

### 2.4. Thrombus Formation in the Flow Chamber (Human, In Vitro)

ASA treatment at concentrations of 1.8 and 9 µg/mL decreased the thrombus area (Figure 4a) and increased the PECAM-1/thrombus ratio (Figure 4b). No statistically significant differences in the thrombus area were observed between groups treated with ASA 1.8 µg/mL and ASA 9 µg/mL. However, ASA administered at a concentration of 9 µg/mL caused a notable increase in the PECAM-1/thrombus ratio in comparison to ASA treatment at 1.8 µg/mL (Figure 4b).

## 3. Discussion

Our study demonstrates that the combination of the PECAM-1/thrombus ratio and flow chamber can be used in animal and human studies. Furthermore, using the lipopolysaccharide (LPS)-induced inflammation model, we observed that the PECAM-1/thrombus ratio calculated in the flow chamber did not differ from that calculated in the intravital model of thrombosis, which suggests that the PECAM-1/thrombus ratio in combination with the flow chamber could be a reliable method for the ex vivo assessment of platelet activity in pathological states.

Based on previous studies, we assume that PECAM-1 expression in a thrombus is related to its expression on platelets since the laser-induced thrombus is largely composed of platelets. This was demonstrated in a previous study, in which the laser-induced thrombus consisted of platelets that had been preloaded with the fluorescent dye FURA 2AM [8]. A model of laser-induced thrombosis has also been used to distinguish platelet populations in the thrombus [9]. Furthermore, it has been shown that under physiological flow conditions, neutrophils are not present in the thrombus within 60 s after laser injury but adhere to activated endothelium downstream of the thrombus site [10]. In the flow chamber model, under the conditions used in this study, the developed thrombus is expected to be composed predominantly of platelets [11]. The contribution of different cells (e.g., leukocytes) could be achieved after specific modifications of the method (significant prolongation of incubation time, the presence of preactivated leukocytes, etc.) [12,13]. Endothelial cells were absent in our flow chamber model.

In studies with rodents, the antiplatelet doses of ASA are higher than doses used in humans: 100 mg/kg i.v. (intravenous) [14], 150 mg/kg i.v. [15], and 200 mg/kg p.o. (per os) [16]. This might stem from the faster metabolism of ASA in rodents (t_1/2_ 0.5–8 min vs. t_1/2_ 20 min in humans) [17,18,19,20] and the higher number of platelets in mice compared to humans (1000−1500 × 10^9^/L vs. 150−400 × 10^9^/L) [21] and, therefore, a larger amount of COX-1 to inhibit.

### 3.1. Mouse Study

In our previous study using the laser-induced thrombosis model, we showed that the increase in the antiplatelet effect correlated with the increase in the PECAM-1/thrombus ratio in healthy mice [4]. In the present study, we confirmed that the PECAM-1/thrombus ratio could also be assessed in the flow chamber model and serve as a valuable parameter to evaluate platelet activity in healthy mice (Figure 2b). Moreover, we evaluated whether the PECAM-1/thrombus ratio could reflect the platelet activation status under pathological conditions, such as LPS-induced inflammation. To ensure that the applied LPS treatment affects hemostasis, we used an experimental protocol in which LPS causes venous and arterial thrombosis in mice (in vivo). The results show that the expression of proinflammatory cytokines, such as tumor necrosis factor α and interleukin 1β, was elevated in thrombus-containing vessels, contributing to enhanced thrombosis in LPS-treated mice [22].

We observed that treatment with LPS alone increased the thrombus area only in the laser-induced thrombosis model (Figure 1a,d), whereas in the flow chamber, the thrombus area remained unchanged (Figure 2c). This indicates that inflamed endothelium most likely enhanced thrombosis under intravital conditions. The curve of thrombus formation kinetics plotted for the LPS group shows attenuated thrombus elution with blood flow, which might suggest enhanced interaction between platelets and the endothelium (Figure 1c). This hypothesis was at least partially confirmed in the experiment in which LPS was found to shorten bleeding time (BT) (Figure 3)—a parameter describing the process of primary hemostasis. BT depends on the blood vessel response, which is mainly regulated by the endothelium (vasoconstriction/vasodilatation), as well as on the activity of platelets that form a platelet plug at the injury site [5]. Our observations are consistent with a previous study that showed that LPS administration enhanced thrombus formation in mouse cremaster microvessels (intravital conditions) and did not intensify platelet aggregation assessed ex vivo [23]. Considering that treatment with LPS alone did not cause any changes in the PECAM-1/thrombus ratio in laser-induced (Figure 1b) and flow chamber models (Figure 2d) or thrombus area (Figure 2c) in the flow chamber, we assume that LPS did not directly affect platelet activity. Due to differences in the experimental protocols, the influence of LPS on platelet activity cannot be clearly defined. However, Martyanov et al. showed that LPS restored the response to ADP in desensitized platelets by activating platelet TLR4 while inhibiting platelet activation by potentiating cAMP/cGMP pathways [24]. This could explain why the most pronounced increase in the PECAM-1/thrombus ratio observed in the LPS + ASA 3 mg/kg group (Figure 1b and Figure 2d) was opposite to the tendency observed in control mice (Figure 2b). This observation was presumably due to a synergistic effect of ASA at a dose of 3 mg/kg and LPS on platelets, which translated into the most potent inhibition. Therefore, in the flow chamber model, a decrease in the thrombus area in the LPS + 3 mg/kg ASA group was observed (Figure 2c). The lack of the antithrombotic effect when ASA was administered at a dose of 3 mg/kg (Figure 1a,d) under intravital conditions, with the most potent increase in the PECAM-1/thrombus ratio (Figure 1b), could be due to the inflammatory phenotype of the endothelium, reflected by shortened BT (Figure 3). The reduced thrombus area observed after treatment with ASA at doses of 10 and 30 mg/kg (Figure 1a,d, and Figure 2c) could be related to its anti-inflammatory effect [25]; thus, higher doses of ASA might have counteracted the synergistic effect of LPS and ASA at a dose of 3 mg/kg on platelet activity. However, in the flow chamber, an increased PECAM-1/thrombus ratio was still observed in groups treated with LPS + ASA 10 mg/kg and LPS + ASA 30 mg/kg in comparison with the LPS group (Figure 2d), which indicates the antiplatelet effect of ASA at all tested doses. The reason for the increased PECAM-1/thrombus ratio along with platelet inhibition after ASA treatment remains to be elucidated. It has been shown that PECAM-1 is shed from the T-cell surface upon proinflammatory activation caused by interaction with antigen-presenting cells [26]. On this basis, it can be assumed that PECAM-1 is shed from platelets during thrombus formation, in which platelet activation and platelet–platelet and platelet–endothelium interactions occur. Another possible reason for the increased PECAM-1/thrombus ratio could be the increased expression of platelet PECAM-1 after ASA treatment and, hence, an increased PECAM-1/thrombus ratio. Therefore, to confirm this hypothesis, future studies using flow cytometry should focus on assessing PECAM-1 expression on platelets that do not form a thrombus.

Taking into account the interactions between platelets in the thrombus and the ability of PECAM-1 to form homophilic complexes [27], we speculate that the PECAM-1/thrombus ratio might characterize platelet aggregation. However, in the flow chamber model, we observed that this parameter did not correlate with the thrombus area in the case of LPS-induced inflammation. Considering the thrombus area as a parameter reflecting the ability of platelets to aggregate on the prothrombotic surface, it can be hypothesized that the PECAM-1/thrombus ratio does not reflect platelet aggregation. Thus, it can be concluded that when used in combination with a flow chamber, the PECAM-1/thrombus ratio can be useful for assessing both platelet aggregation (thrombus area) and platelet activity (PECAM-1/thrombus ratio).

In both models of thrombosis, platelets interact with collagen. However, puncturing the vessel wall by the ablation laser used in our study exposed a smaller area of collagen compared with the collagen area in the flow chamber. This makes the laser-induced thrombus less stable and prone to elution with the blood flow [4]. This allows us to observe a rapid platelet response to vessel wall injury. In a laser-induced thrombosis platelet, secretion occurs to a limited degree only at the site of vessel wall injury [5], whereas in the flow chamber model, we observed firm adhesion to the collagen surface, more stable thrombi, and potent platelet secretion within the whole thrombus [28]. Despite the above, the direction of change in the PECAM-1/thrombus ratio was convergent regardless of prothrombotic conditions, and we were able to assess the PECAM-1/thrombus ratio in both models of thrombosis. Therefore, we conclude that the PECAM-1/thrombus ratio can be assessed in conditions associated with a different platelet activation state.

We cannot state the cause of the difference in tail-bleeding times between the 10 and 30 mg/kg ASA groups. A dose-independent effect of ASA on the endothelial inflammatory response in LPS-treated mice has been reported [29]. However, in the cited study, ASA was administrated to mice for 7 days, whereas in our study, ASA was administrated once, 5 min before thrombosis induction or blood collection.

### 3.2. Human Study

The assessment of platelet activity is necessary for a variety of clinical settings [30]. Therefore, in this study, we performed a preliminary in vitro experiment in a flow chamber to verify whether the PECAM-1/thrombus ratio is also applicable to human blood. For this purpose, we used ASA in clinically relevant concentrations determined after its administration at antiplatelet doses [31,32].

A clinical study showed that administering clopidogrel with ASA reduced the expression of platelet PECAM-1 in patients who had suffered an ischemic stroke. As an adhesion molecule, PECAM-1 was identified as an indicator of increased platelet activity, and, thus, decreased expression of platelet PECAM-1 has been linked to the antiplatelet effect [33]. On the contrary, our study showed that an increased PECAM-1/thrombus ratio in the thrombus correlated with decreased platelet activity (Figure 4). This difference in findings is most likely due to the fact that, in the abovementioned clinical study [33], the expression of platelet PECAM-1 was assessed using flow cytometry, in which platelet interactions are negligible. This suggests that PECAM-1 expression restricted only to resting platelets should not be considered a marker of platelet activity. Furthermore, PECAM-1 on nonactivated circulating blood cells cannot form homophilic complexes [27], which might indicate that, during thrombosis, PECAM-1 functions as a feedback inhibitor of thrombus formation.

Our study showed no difference in thrombus area between treatments with ASA 9 µg/mL and ASA 1.8 µg/mL, but the PECAM-1/thrombus ratio observed with ASA 9 µg/mL was significantly higher in comparison with ASA 1.8 µg/mL. Therefore, we suggest that compared with the thrombus area, the PECAM-1/thrombus ratio can be a more precise parameter for describing platelet activity, as concluded based on the mouse study.

The novelty of our method is that only platelets forming the thrombus are assessed for activity, which is more physiologically relevant than platelet activation markers measured by flow cytometry in a platelet suspension. Furthermore, this method is a promising tool for the screening of cardiovascular risk associated with altered platelet activity. Measuring the PECAM-1/thrombus ratio could also be a diagnostic strategy for identifying platelet-related hemostasis abnormalities. Changes in the PECAM-1/thrombus ratio in human blood after ASA treatment indicate the possibility of applying our method to monitor the efficiency of antiplatelet pharmacotherapy. Using a laser model of thrombosis, we also showed the ability to assess platelet activity while simultaneously evaluating the thrombotic process under intravital conditions. This is particularly useful in the comprehensive evaluation of the mechanism of action of new substances with potential antithrombotic activity.

In summary, our study shows that the platelet PECAM-1 molecule can be used for the assessment of platelet activation in both normal conditions and pathological states such as inflammation. Using the LPS-induced inflammation model, we demonstrate for the first time that the PECAM-1/thrombus ratio more accurately describes the platelet activation status than commonly used parameters such as the thrombus area. Furthermore, we show for the first time that the PECAM-1/thrombus ratio can also be applied to human blood.

## 4. Materials and Methods

### 4.1. Blood Collection for Human Study

The study protocol was reviewed and accepted by the Bioethics Committee of the Medical University of Bialystok (Approval No.: APK.002.283.2021, date of approval: 27 May 2021). After obtaining informed consent, in accordance with the Declaration of Helsinki, blood samples were collected from healthy volunteers in vacutainers containing K_2_EDTA (1.8 mg/mL) as an anticoagulant.

### 4.2. Animals

Male wild-type *C57BL/6*J mice (weighing 25–27 g) were used in the experiments. These experiments were conducted in accordance with the EU Guidelines on Animal Experiments (European Directive 2010/63/EU), and all procedures involving animals and their care were approved by the Local Ethical Committee on Animal Testing in Olsztyn (Approval No.: 93/2018, date of approval: 18 December 2018). The animals were purchased from the Center for Experimental Medicine in Bialystok. They were anesthetized by intraperitoneally (i.p.) administering a single dose of ketamine (120 mg/kg, Ketamina 10%; Biowet, Poland) and xylazine mixture (12.5 mg/kg, Xylapan; Biowet, Poland). Blood samples were collected from the right heart ventricle of mice with a 3.8% sodium citrate solution (1:10, *v*/*v*). After the experiments, the animals were killed by cervical dislocation.

### 4.3. Induction of Inflammation

Inflammation was induced using a procedure described by Wang (2008) [22]. Briefly, LPS (from *Escherichia coli* Serotype 026:B6, 2 mg/kg; Sigma-Aldrich, Darmstadt, Germany) prepared in saline was administered to mice by a single i.p. injection. Mice without inflammation were administered (i.p.) only saline. For the flow chamber experiment, blood samples were collected from the animals 30 min after LPS injection. In the laser-induced thrombosis model, thrombus formation was induced 30 min after LPS injection.

### 4.4. ASA Administration

A 0.1 mL volume of ASA in phosphate-buffered saline (PBS) (Flectadol 1 g/5 mL; Sanofi, Paris, France) or 0.1 mL of PBS was injected into the left femoral vein in mice 5 min before blood collection or thrombosis induction. In the experiment with the flow chamber, mice that received ASA without LPS treatment were referred to as control mice. The doses of ASA (3, 10, or 30 mg/kg) were chosen based on our previous study [4].

The concentrations of ASA (0.18, 1.8, 9 µg/mL) used in the experiment with human blood were clinically relevant and reflected the blood concentration observed after the administration of antiplatelet doses of ASA [31,32].

### 4.5. Microscopic Visualization

A fixed-stage Zeiss Axio Examiner Z.1 microscope (Carl Zeiss Microscopy GmbH, Germany) equipped with a confocal scanning unit (Yokogawa CSU-X1; Yokogawa Electric Corporation, Japan) was used for all microscopic analyses. Confocal images were captured using a high-speed digital camera (CCD C9300-221; Hamamatsu Photonics K.K., Hamamatsu City, Japan).

### 4.6. Flow Chamber Characteristics and Preparation of Collagen-Coated Surfaces

The characteristics of the flow chamber used in this study are described in our previous paper [4]. The flow chamber used was a transparent, polycarbonate-made, parallel-plate type (height, 50 μm; width, 3 mm; length, 30 mm) and was equipped with a metal (medical steel) inlet and an outlet installed at an angle of 15° in relation to the flow axis. Disposable coverslips (0.2 mm thick, degreased in a mixture of 2 M HCl and 50% ethanol) were used to create a detachable bottom for the chamber. To facilitate the formation of the thrombus under flow conditions, the coverslips were coated with a microspot of type I collagen suspension (diluted with 1.67 mM acetic acid to a concentration of 50 μg/mL, Horm collagen; Nycomed, Zurich, Switzerland). The spot had a diameter of 1 mm. After overnight incubation at 4 °C in a humid chamber, the coverslips were washed with saline (to remove unbound collagen) and blocked for 30 min with 1% (*w*/*v*) bovine serum albumin (BSA) in Hepes buffer (138 mM NaCl, 2.8 mM KCl, 8.9 mM NaHCO_3_, 0.8 mM KH_2_PO_4_, 0.8 mM MgCl_2_, 5.5 mM glucose, 3.5 mg/mL albumin, and 10 mM HEPES, pH 7.4) in order to minimize nonspecific interactions of blood constituents with glass. After blocking, the coverslips were rinsed again with saline before the experiment. Then, the chamber was assembled and filled with Hepes buffer containing 0.1% BSA and glucose (5.5 mM).

### 4.7. Model of Thrombus Formation in a Flow Chamber

To inhibit coagulation during the course of the assay, blood was treated with d-phenylalanyl-l-prolyl-l-arginine chloromethyl ketone (PPACK) (final concentration, 40 µM; Santa Cruz Biotech., Dallas, TX, USA), a selective, irreversible thrombin inhibitor. Then, the samples were supplemented with 3,3′-dihexyloxacarbocyanine iodide (DiOC6(3), 1 µM; Life Technologies, Molecular Probes, USA) and incubated for 2 min at 37 °C to stain platelets. For the human study, the blood samples were supplemented with ASA solution (final concentration, 1, 10, or 50 µM) and incubated for 3 min. Shortly before the perfusion of blood into the chamber, the samples were supplemented with 10 mM CaCl_2_ and 3.37 mM MgCl_2_ (for mouse blood, both at 3 mM concentration). Whole blood was perfused into the chamber via silicon tubing (inner diameter, 1 mm). The flow rate was set such that the wall shear rate was 1000 s^−1^ according to the Poiseuille equation [34,35]. The shear rate reflected the conditions of arterial circulation. For the full formation of the thrombus on the collagen-coated surface, blood flow through the chamber was continued for 4 min (starting from the entry into the chamber), and then the thrombus-covered surface was washed with Hepes buffer. After disassembling the flow chamber, PECAM-1 within thrombi was stained by topically applying 2 µL of Alexa Fluor 647-labeled PECAM-1 antibody (Alexa Fluor 647 anti-mouse CD31 antibody or Alexa Fluor 647 anti-human CD31 antibody; Bio Legend, San Diego, CA, USA) on a thrombi-rich area. Then, the samples were incubated with anti-CD31 antibody at room temperature for 5 min and washed out, which ensured that we observed PECAM-1 present on the cell surface. Furthermore, according to the manufacturer’s information, the anti-CD31 antibody was used for immunohistochemical staining of PECAM-1 present on the cell surface, which excludes the possibility of staining the soluble form of PECAM-1.

In the formed thrombi, end-stage measurements (including the surface area covered by the thrombus, referred to as the thrombus area) were performed by taking two-color fluorescent pictures using a confocal microscope. Further analysis was carried out using the SlideBook 6.0 software (Intelligent Imaging Innovation, Inc., Denver, CO, USA). The area of fluorescence from platelet PECAM-1 was divided by the thrombus area, and the value thus obtained was referred to as the PECAM-1/thrombus ratio.

### 4.8. Laser-Induced Thrombosis in Mouse Mesenteric Vein and Assessment of Thrombus Area and PECAM-1/Thrombus Ratio (Intravital)

Laser induction of thrombosis was performed according to our previous study with some modifications [5]. Five minutes before the mesentery vein wall damage, Alexa Fluor 647-labeled PECAM-1 antibody (0.25 mg/kg, Alexa Fluor 647 anti-mouse CD31 antibody; Bio Legend, San Diego, CA, USA) was injected into the femoral vein of mice. To visualize the thrombus, DiOC6(3) (0.1 mM in 0.05 mL of dimethyl sulfoxide and PBS mixture (volume ratio, 1:50); Life Technologies, Molecular Probes, Waltham, MA, USA) was intramuscularly administered. A midline laparotomy incision was made, and the mesentery of the ileum was pulled out of the abdomen and draped over a plastic mound. The mesentery vein was microscopically identified and continuously perfused with prewarmed (37 °C) PBS to prevent the vessels from drying. Then, the vein wall was injured by a 532 nm argon-ion ablation laser (Ablate™; Intelligent Imaging Innovations, Inc., Denver, CO, USA). The induction and progression of thrombosis were recorded for 80 s. One record was divided into 25 time points, and at each time point, the thrombus area was encircled. The thrombus area values obtained at each time point were added to estimate the final value. In addition, the activity of platelets in the thrombus was assessed by measuring the area of fluorescence of PECAM-1 at each time point. The area of PECAM-1 fluorescence determined at a particular time point was then divided by the thrombus area at that time point. The values from one record were added to estimate the PECAM-1/thrombus ratio. One thrombus was induced in each mouse.

### 4.9. Determination of BT in Mice

BT analysis was performed in anesthetized mice as described by Vital et al. (2020). Briefly, a small tail segment (3 mm) was cleanly cut with a scalpel blade. Bleeding was monitored at 15-min intervals by absorbing the bead of blood with filter paper without contacting the wound site. When no blood was observed on the paper, it was assumed that bleeding had stopped [36].

### 4.10. Statistical Analysis

The obtained data were analyzed using GraphPad Prism 5. The normality of the data distribution was assessed using the Shapiro–Wilk test. Differences between groups were assessed using analysis of variance (for normally distributed data) or the Kruskal–Wallis test (for non-normally distributed data) with the appropriate post hoc test. The results are expressed as mean ± SEM (for normally distributed data) or as median (with interquartile range; for non-normally distributed data) of the number of determinations (n). A *p*-value of <0.05 was considered significant.

## Figures and Tables

**Figure 1 ijms-22-09611-f001:**
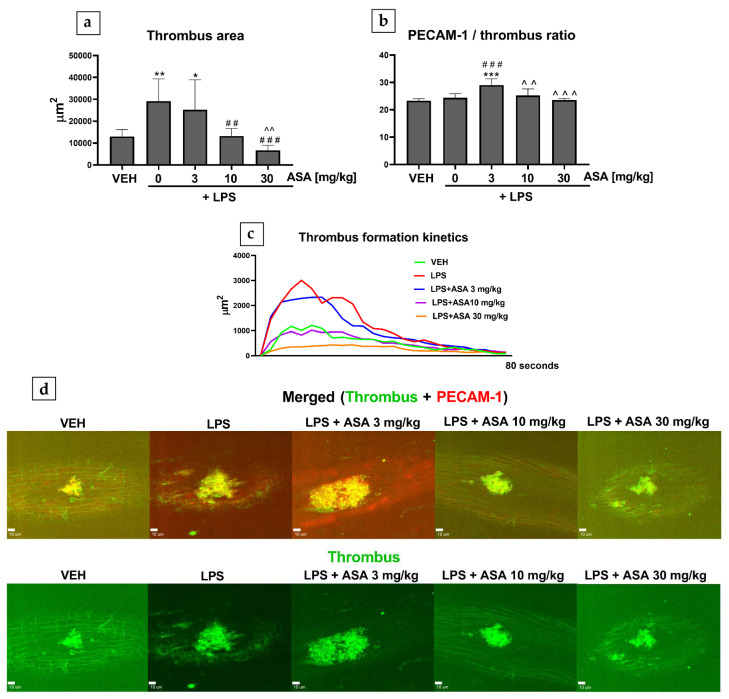
Effect of ASA (intravital) on (**a**) thrombus area and (**b**) PECAM-1/thrombus ratio in mouse mesentery vein. (**c**) Kinetics of thrombus formation at the site of laser injury. (**d**) Original, representative confocal microscopy images of merged channels (green, thrombus; red, PECAM-1) and thrombus (green) 10 s after thrombosis induction (bar = 10 µm; * *p* < 0.05, ** *p* < 0.01, *** *p* < 0.001 vs. VEH; ^##^
*p* < 0.01, ^###^
*p* < 0.001 vs. LPS; ^^ *p* < 0.01, ^^^ *p* < 0.001 vs. LPS + ASA 3 mg/kg; *n* = 7−8). Data are shown as mean ± SEM.

**Figure 2 ijms-22-09611-f002:**
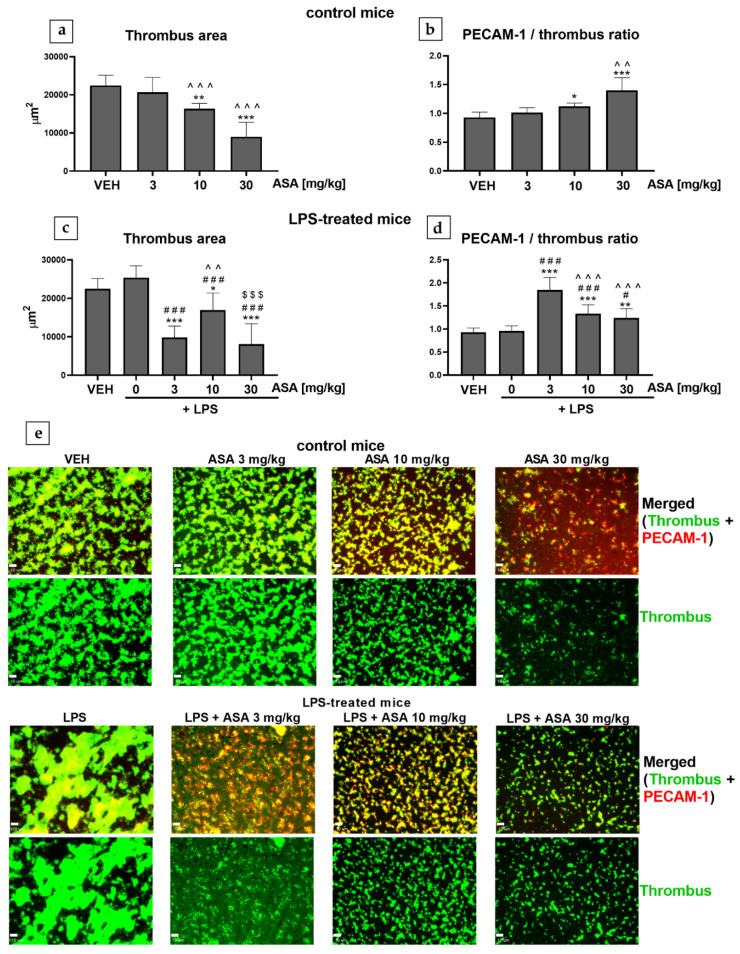
Effect of ASA (ex vivo) on (**a**) thrombus area and (**b**) PECAM-1/thrombus ratio in mice without inflammation (control mice) and on (**c**) thrombus area and (**d**) PECAM-1/thrombus ratio in mice with inflammation (LPS-treated mice). (**e**) Top panel, control mice; bottom panel, LPS-treated mice; original, representative confocal microscopy images of merged channels (green, thrombus; red, PECAM-1, top row in each panel) and thrombus (green, bottom row in each panel) (bar = 10 µm; * *p* < 0.05, ** *p* < 0.01, *** *p* < 0.001 vs. VEH; ^#^
*p* < 0.05, ^###^
*p* < 0.001 vs. LPS; ^^ *p* < 0.01, ^^^ *p* < 0.001 vs. LPS + ASA 3 mg/kg; ^$$$^
*p* < 0.001 vs. LPS + ASA 10 mg/kg; *n* = 9−13). Data are shown as mean ± SEM.

**Figure 3 ijms-22-09611-f003:**
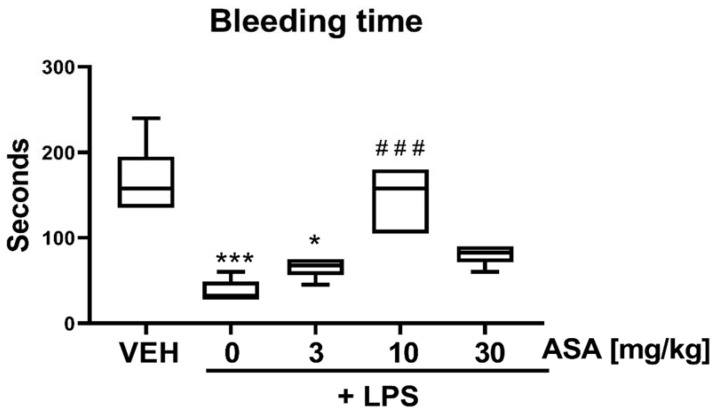
Effect of ASA on BT in LPS-treated mice (* *p* < 0.05, *** *p* < 0.001 vs. VEH; ^###^
*p* < 0.001 vs. LPS; *n* = 6). Data are shown as median (interquartile range).

**Figure 4 ijms-22-09611-f004:**
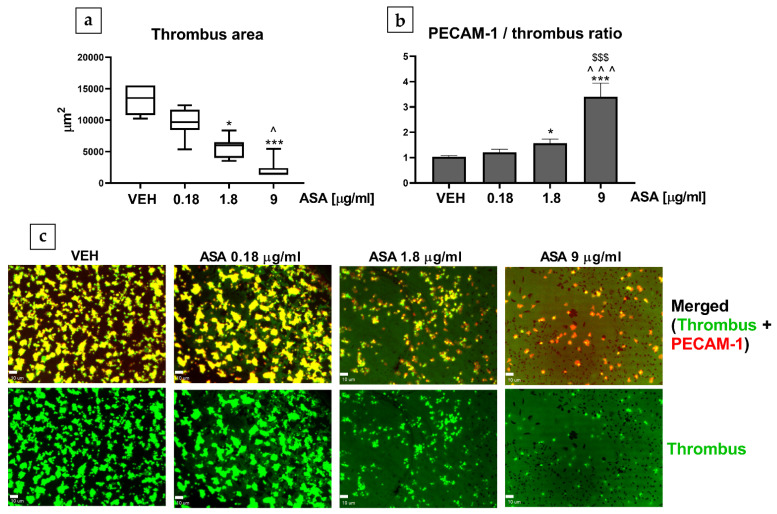
Effect of ASA (in vitro) on (**a**) thrombus area and (**b**) PECAM-1/thrombus ratio in the experiment with human blood. (**c**) Original, representative confocal microscopy images of merged channels (green, thrombus; red, PECAM-1) and thrombus (green) (bar = 10 µm; * *p* < 0.05, *** *p* < 0.001 vs. VEH; ^ *p* < 0.05, ^^^ *p* < 0.001 vs. ASA 0.18 µg/mL; ^$$$^
*p* < 0.001 vs. ASA 1.8 µg/mL; *n* = 7). Data are shown as mean ± SEM and as median (interquartile range).

## Data Availability

The data presented in this study are available on request from the corresponding author.

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
