# Peer review of "Utility of Platelet Endothelial Cell Adhesion Molecule 1 in the Platelet Activity Assessment in Mouse and Human Blood"

_ijms, 2021, doi:10.3390/ijms22179611_

Round 1

Reviewer 1 Report

The manuscript of Marcinczyk et al covers an important aspect of thrombo-inflammation. The methods seem generally up-to-date, but several points require attention:

Major comments:

  1. The PECAM-1 molecule was identified by an Alexa-Fluor647 -labeled PECAM-1 antibody. The Alexa fluorophors have the following emission color: 488: green, 594: red, 647: magenta. How come that the 647 fluorophor is shown up in red on the slides ?
  2. PECAM-1 is expressed on several cell types, platelets, leukocytes, and endothelial cells. How can the authors exclude that the observed elevated PECAM-1 expression is not related to cells other than platelets. This is particularly plausible in case of the LPS experiments, since monocytes react abruptly to LPS stimulation.
  3. Also it should be specified whether the PECAM-1 staining is only cell-related or does it also detect the soluble form of PECAM-1 ?
  4. The figure sizes in general does not allow proper identification of the proposed labels i.e. no bar is visible on Figure 4, on Figure 3 the Y axis only show 'S' that is probably Tail bleeding time in seconds and should be properly indicated.
  5. What causes the significant difference in tail-bleeding times between the 10 and 30 mg/kg ASA groups ?
  6. The authors should unify the mouse and human studies regarding the expression of the ASA doses. Primarily the 3-30 mg/kg should be justified in mice and should be compared to humans and explain why is that significantly different (a 70-80 kg human takes a 100 mg ASA tablet). Secondly in human studies ASA dose is expressed in molarity and not mass units. These should be unified.

Minor comments:

  • English requires some further attention  e.g. the sentence in lines 60-62 is grammatically incorrect
  • The 'Shp' acronym should be defined when first used

Author Response

  1. The PECAM-1 molecule was identified by an Alexa-Fluor647 -labeled PECAM-1 antibody. The Alexa fluorophors have the following emission color: 488: green, 594: red, 647: magenta. How come that the 647 fluorophor is shown up in red on the slides ?

According to manufacturer’s (Thermo Fisher Scientific) information Alexa 647 is a fluorescent dye with excitation peak at 650 nm and emission peak at 665 nm (range 640-760 nm, with max. at  671 nm, as shown in the excitation-emission spectrum: https://www.thermofisher.com/order/fluorescence-spectraviewer?SID=srch-svtool&UID=21235p72#!/ ). Emission waves emitted by Alexa 647 passes through the appropriate filter and are detected by the CCD camera. Due to the construction of CCD matrice, the camera detect wave emitted by Alexa 647 and create monochrome image because the detection of emission wave by the CCD camera is not associated with the information about colour of this wave. The red color shown on slide is the matter of the software (SlideBook 5.0) settings that displays images captured by the camera.  

  1. PECAM-1 is expressed on several cell types, platelets, leukocytes, and endothelial cells. How can the authors exclude that the observed elevated PECAM-1 expression is not related to cells other than platelets. This is particularly plausible in case of the LPS experiments, since monocytes react abruptly to LPS stimulation.

Thank you for pointing out this important issue. We did not perform the morphological assessment of thrombus. However, based on the previous studies we assume that PECAM-1 expression in thrombus was related to its expression on platelets since laser-induced thrombus formed in this model is composed in a vast majority of platelets. It was demonstrated in study where laser-induced thrombus was composed of platelets that had been previously preloaded with fluorescent dye FURA 2AM (Dubois C et al. 2007, Thrombin-initiated platelet activation in vivo is vWF independent during thrombus formation in a laser injury model). Model of laser-induced thrombosis is used also to distinguish the platelet populations in thrombus (Hayashi T et al. 2008, Real-time analysis of platelet aggregation and procoagulant activity during thrombus formation in vivo). Furthermore, it has been shown that under physiological flow within 60 seconds after laser injury neutrophils are not present in thrombus but adhere to activated endothelium downstream of thrombus site (Gollomp K et al. 2016, A special role for neutrophil extracellular traps (NETs) and neutrophils in the prothombotic nature of heparin-induced thrombocytopenia). In the flow chamber model, under conditions we used, development of thrombus composed predominantly of platelets is expected (de Witt S et al. 2014, Identification of platelet function defects by multi-parameter assessment of thrombus formation). The contribution of different cells (e.g. leukocytes) could be achieved after specific modification of method (significant prolongation of incubation time, the presence of preactivated leukocytes, etc. [Clark SR et al. 2007, Platelet TLR4 activates neutrophil extracellular traps to ensnare bacteria in septic blood, Frydman et al. 2017, Technical Advance: Changes in neutrophil migration patterns upon contact with platelets in a microfluidic assay). Endothelial cells were absent in our flow chamber model.

Pages 7-8, lines: 154-166.

  1. Also it should be specified whether the PECAM-1 staining is only cell-related or does it also detect the soluble form of PECAM-1 ?

According to manufacturer’s information the anti-CD31 antibody is used to immunohistochemical staining of PECAM-1 present on the cell surface. Furthermore, thrombi obtain in the flow chamber model were washed out which ensures that we observed PECAM-1 present on the cell surface.

Page 12, lines: 380-383.

  1. The figure sizes in general does not allow proper identification of the proposed labels i.e. no bar is visible on Figure 4, on Figure 3 the Y axis only show 'S' that is probably Tail bleeding time in seconds and should be properly indicated.

To obtain better quality of figures, the labels and resolution were changed. We hope that improved resolution and optimal pixel density should result in better quality of graphs. Y axis on Figure 3 (page 6) has been modified. New Figures have been inserted into text.

  1. What causes the significant difference in tail-bleeding times between the 10 and 30 mg/kg ASA groups?

We cannot state what was the cause of difference in tail-bleeding times between the 10 and 30 mg/kg ASA groups. Dose-independent effect of ASA on endothelial inflammatory response in LPS-treated mice was reported (Zhou et al. 2019, Aspirin alleviates endothelial gap junction dysfunction through inhibition of NLRP3 inflammasome activation in LPS-induced vascular injury). However, in above study ASA was administrated to mice for 7 days whereas in our study ASA was administrated once, 5 minutes before thrombosis induction or blood collection.

Page 9, lines: 254-258.

We plan to investigate this phenomenon in future studies. However, this difference was not statistically significant.

  1. The authors should unify the mouse and human studies regarding the expression of the ASA doses. Primarily the 3-30 mg/kg should be justified in mice and should be compared to humans and explain why is that significantly different (a 70-80 kg human takes a 100 mg ASA tablet). Secondly in human studies ASA dose is expressed in molarity and not mass units. These should be unified.

Thank you for pointing this out. In studies with rodent the antiplatelet doses of ASA are higher than doses used in humans: 100 mg/kg i.v. (Lorrain J et al. 2004 , Effects of SanOrg123781A, a synthetic hexadecasaccharide, in a mouse model of electrically induced carotid artery injury: synergism with the antiplatelet agent clopidogrel), 150 mg/kg i.v. (Aktas B at al. 2005, Aspirin induces platelet receptor shedding via ADAM17 (TACE)), 200 mg/kg p.o. (Lee et al. 2021, Safety and efficacy of targeting platelet proteinase-activated receptors in combination with existing anti-platelet drugs as antithrombotics in mice). This might stem from faster metabolism of ASA in rodents (t1/2 0.5–8 min vs t1/2 20 min in human) (Wientjes M. and Levy G. 1988, Nonlinear pharmacokinetics of aspirin in rats, Fu CJ et al. 1991, The pharmacokinetics of aspirin in rats and the effect of buffer, Nagelschmitz et al. 2014, Pharmacokinetics and pharmacodynamics of acetylsalicylic acid after intravenous and oral administration to healthy volunteers, Droebner et al. 2017, Pharmacodynamics, pharmacokinetics, and antiviral activity of BAY 81-8781, a novel NF-κB inhibiting anti-influenza drug) and higher number of platelets in mice compared with human (1000–1500 × 109/L vs 150–400 × 109/L) (Schmitt at al. 2001, Of mice and men: comparison of the ultrastructure of megakaryocytes and platelets) and therefore larger amount of COX-1 to inhibit.

Page 8, lines: 167-171.

The molarity units have been changed to mass units.

Minor comments

  1. Thank you for pointing this out. It has been changed in the revised version of the manuscript (page 2, line 60). Language verification of the manuscript was performed. Please find attached language editing certificate.
  2. “Shp” acronym has been defined. Page 2, line 49-50.

Reviewer 2 Report

In this paper, the authors aim to show that PECAM-1/thrombus ratio is a useful measure of to assess platelet activation. The study used both mice (intravital and ex vivo experiments) and human blood to assess the hypothesis. The authors conclude that PECAM-1 can be used for assessment of platelet activation and that the PECAM-1/thrombus ratio can be applied to both mouse and human blood.

Comments

My main concern is the overall applicability of this method as an assessment of platelet activity. There seems to be a presumption that thrombus formation is exclusively dependent on platelet activation, but there is now abundant evidence that other processes such as NETosis and monocyte ETosis are key contributors to thrombus formation in several pathological states (not necessarily in LPS-induced inflammation). For the PECAM-1/thrombus ratio to be adopted the authors might need to show that it applies more widely.

Discussion, lines 203-204: the manuscript states “Another possible reason for increased PECAM-1/thrombus ratio could be increased expression of platelet PECAM-1 after ASA treatment and hence increased PECAM-1/thrombus ratio”. The authors should test this experimentally by determining PECAM-1 expression on platelets by flow cytometry in the presence and absence of ASA (both in mouse and human platelets).

The discussion argues that in the vessel injury model, platelets did not interact effectively with collagen, while in the flow chamber collagen was used to coat the channel thus providing an extensive procoagulant surface. Based on this the authors conclude that PECAM-1 as a molecule regulating thrombotic process regardless of the initial prothrombotic trigger. I think that more evidence is needed to make this statement. Have the authors shown that in the vessel injury model platelets do no interact effectively with collagen? If they don’t interact with collagen, what is the reason for platelet activation and thrombus formation following laser injury?

The discussion states (line 194) that the higher doses of ASA might have attenuated the LPS-inhibitory effect on platelets. The meaning of this sentence is not clear to this reviewer.

Other comments

For consistency Figures 2b and 2d should be in the same format

LPS-untreated and LPS-treated are somewhat confusing to the reader. It might be better to say something like control mice and LPS-treated mice.

P2, line 60, please change “the less are the activated platelets in the thrombus” to “the less activated platelets are present in the thrombus”

P2, line 65, please change “activity of human platelets in ex vivo study” to “activity of human platelets ex vivo”

P2, line 79, please change “two models is the presence or lack of endothelium” to “two models is the presence or absence of endothelium”

Figures 1c and 1d are not mentioned in the text

Author Response

  1. My main concern is the overall applicability of this method as an assessment of platelet activity. There seems to be a presumption that thrombus formation is exclusively dependent on platelet activation, but there is now abundant evidence that other processes such as NETosis and monocyte ETosis are key contributors to thrombus formation in several pathological states (not necessarily in LPS-induced inflammation). For the PECAM-1/thrombus ratio to be adopted the authors might need to show that it applies more widely.

The novelty of our method may be proved by the fact that only platelets forming thrombus are assessed for activity which is more physiologically relevant than the measurement of the platelet activation markers by flow cytometry in a platelet suspension. Furthermore, this method is a promising tool for the screening of cardiovascular risk associated with altered platelet activity. PECAM-1/thrombus ratio could be also one of the diagnostic strategy for identifying platelet-related hemostasis abnormalities. Changes in PECAM-1/thrombus ratio in human blood after ASA treatment initially indicate the possibility of applying our method to monitor the efficiency of antiplatelet pharmacotherapy. Using laser model of thrombosis we showed also the ability of platelet activity assessment with simultaneous evaluation of the thrombotic process under intravital condition. It is particularly useful in comprehensive evaluation of the mechanism of action of new substances with potential antithrombotic activity. Page 10, lines: 283-294.

We agree that thrombus formation is a complex process which involves multiple types of cells. We are aware that in different pathological states (inflammation, diabetes, hypertension) the functions of hemostasis components is altered and could affect thrombus by different mechanisms. However, the overriding goal of our study was to find a parameter that would allow only assessment of platelet activity. Page 2, lines 82-83 .

We suppose that contribution of NET in our experimental model is negligible since their formation is a late response (hours) compared with platelet thrombus formation in our study (seconds) (Masuda et al. 2021, Measurement of NET formation in vitro and in vivo by flow cytometry).  

  1. Discussion, lines 203-204: the manuscript states “Another possible reason for increased PECAM-1/thrombus ratio could be increased expression of platelet PECAM-1 after ASA treatment and hence increased PECAM-1/thrombus ratio”. The authors should test this experimentally by determining PECAM-1 expression on platelets by flow cytometry in the presence and absence of ASA (both in mouse and human platelets).

We agree with the reviewer’s assessment and stated this briefly in the previous version of manuscript. This aspect was highlighted in the revised version (Page 9, lines: 229-230). Our hypothesis needs to be confirmed in the future. The assessment of PECAM-1 expression with flow cytometry is planned.

  1. The discussion argues that in the vessel injury model, platelets did not interact effectively with collagen, while in the flow chamber collagen was used to coat the channel thus providing an extensive procoagulant surface. Based on this the authors conclude that PECAM-1 as a molecule regulating thrombotic process regardless of the initial prothrombotic trigger. I think that more evidence is needed to make this statement. Have the authors shown that in the vessel injury model platelets do no interact effectively with collagen? If they don’t interact with collagen, what is the reason for platelet activation and thrombus formation following laser injury?

Thank you for pointing this out. We should have described this issue more clearly. In both model of thrombosis platelets interact with collagen. However, puncture of the vessel wall by the ablation laser used in our study exposes a small area of collagen compared with collagen area in the flow chamber. It makes laser-induced thrombus less stable and prone to be eluted with the blood flow (Marcinczyk N et al. 2020, New approaches for the assessment of platelet activation status in thrombus under flow condition using confocal microscopy.). This allows to observe rapid platelet response to vessel wall injury. In a laser-induced thrombosis platelet secretion occurs in a limited degree only at the site of wall vessel injury (Marcinczyk N et al. 2021, Multidirectional effects of tormentil extract on hemostasis in experimental diabetes) whereas in the flow chamber model we observed firm adhesion to collagen surface, more stable thrombi, and a potent platelet secretion within the whole thrombus (Misztal T et al. HAuCl4, putative general aquaporins blocker, reduces platelet spreading, filopodia formation, procoagulant response, and thrombus formation under flow). Despite the above, the direction of changes in the PECAM-1/thrombus ratio was converged regardless of prothrombotic conditions and we were able to assess PECAM-1/thrombus ratio in both models of thrombosis. Therefore, we concluded that PECAM-1/thrombus ratio can be assessed in conditions associated with a different platelet activation state. Page 9, lines: 242-253.

  1. The discussion states (line 194) that the higher doses of ASA might have attenuated the LPS-inhibitory effect on platelets. The meaning of this sentence is not clear to this reviewer.

The sentence was poorly expressed. Thank you for pointing this out. LPS alone did not decrease platelet activity. However, the synergistic effect of ASA at the dose of 3 mg/kg and LPS could lead to potent platelet inhibition which was probably counteracted by the higher doses of ASA.

Pages 8-9, lines: 208-218.

Other comments

  1. For consistency Figures 2b and 2d should be in the same format

The format of the Figure 2b has been changed. Page 5.

  1. LPS-untreated and LPS-treated are somewhat confusing to the reader. It might be better to say something like control mice and LPS-treated mice.

“LPS-untreated mice” has been changed to “control mice”.

  1. P2, line 60, please change “the less are the activated platelets in the thrombus” to “the less activated platelets are present in the thrombus”

It has been changed. Page 2, line 60.

  1. P2, line 65, please change “activity of human platelets in ex vivo study” to “activity of human platelets ex vivo”

It has been changed. Page 2, line 65.

  1. P2, line 79, please change “two models is the presence or lack of endothelium” to “two models is the presence or absence of endothelium”

It has been changed. Page 2, line: 80.

  1. Figures 1c and 1d are not mentioned in the text

Figure 1c was mentioned in the previous version of the manuscript (in revised version page 8, line: 191). Figure 1d is mentioned in the revised version (page 8, lines: 186, 212, 216).

Round 2

Reviewer 1 Report

I accept the revised version that is now considerably better.

Author Response

The manuscript has undergone english editing. Please see the attachment.

Reviewer 2 Report

I have reviewed the revised manuscript by Marcinczyk et al.

I'm satisfied with the authors' reply to my previous comments. I believe that the manuscript is now ready for publication after suitable review of English-language style.

Author Response

(The authors gave the same response as above.)
